# Initiation and Execution of Programmed Cell Death and Regulation of Reactive Oxygen Species in Plants

**DOI:** 10.3390/ijms222312942

**Published:** 2021-11-30

**Authors:** Chanjuan Ye, Shaoyan Zheng, Dagang Jiang, Jingqin Lu, Zongna Huang, Zhenlan Liu, Hai Zhou, Chuxiong Zhuang, Jing Li

**Affiliations:** 1State Key Laboratory for Conservation and Utilization of Subtropical Agro-Bioresources, College of Life Sciences, South China Agricultural University, Guangzhou 510642, China; chanjuanye@163.com (C.Y.); zhengsy1229@163.com (S.Z.); dagangj@scau.edu.cn (D.J.); ljq110328@163.com (J.L.); hzn20203164101@163.com (Z.H.); zhenlan_liu@scau.edu.cn (Z.L.); hai_zhou127@126.com (H.Z.); zhuangcx@scau.edu.cn (C.Z.); 2Key Laboratory of Plant Functional Genomics and Biotechnology of Guangdong Provincial Higher Education Institutions, South China Agricultural University, Guangzhou 510642, China; 3College of Life Sciences, South China Agricultural University, Guangzhou 510642, China

**Keywords:** PCD, initiation, execution, ROS, regulation

## Abstract

Programmed cell death (PCD) plays crucial roles in plant development and defence response. Reactive oxygen species (ROS) are produced during normal plant growth, and high ROS concentrations can change the antioxidant status of cells, leading to spontaneous cell death. In addition, ROS function as signalling molecules to improve plant stress tolerance, and they induce PCD under different conditions. This review describes the mechanisms underlying plant PCD, the key functions of mitochondria and chloroplasts in PCD, and the relationship between mitochondria and chloroplasts during PCD. Additionally, the review discusses the factors that regulate PCD. Most importantly, in this review, we summarise the sites of production of ROS and discuss the roles of ROS that not only trigger multiple signalling pathways leading to PCD but also participate in the execution of PCD, highlighting the importance of ROS in PCD.

## 1. Introduction

Programmed cell death (PCD) is a genetically controlled cell death process that is activated and executed by the cell itself and is commonly observed in both eukaryotic and prokaryotic organisms [1,2]. In plants, PCD is necessary for growth and survival and is used to eliminate superfluous or damaged cells [2,3], thereby playing a central role in plant development as well as in inducing defence response and the response to environmental conditions.

Three types of PCD have been recognised in animals: apoptosis, autophagic cell death, and necrosis [4]. Plant PCD can be divided into two types based on the function, namely development-related PCD (dPCD) and environment-related PCD (ePCD). Subcellular structures such as mitochondria, endoplasmic reticulum, and nucleus are involved in animal PCD. Chloroplasts and vacuoles are also involved in plant PCD [2].

dPCD plays a regulatory role in cell differentiation, biological development, and senescence of organs [5,6]. dPCD can occur locally or on a wide scale in physiological processes of plants such as degradation of synergids, degeneration of endosperm cells, degradation of the aleurone layer, rupture of the seed coat, formation of vascular vessels, senescence and abscission of leaves, and degradation of megaspores and anther wall cells [3,7,8,9,10,11,12,13]. dPCD is driven by intracellular and extracellular signals. The cell must perceive and evaluate these signals, triggering signal transduction cascades. Afterwards, the cell decides whether to act upon the signals to execute cell death [6]. Plant ePCD is induced by biotic and abiotic stresses, such as infection by pathogenic microorganisms [3,14], ultraviolet radiation, and accumulation of peroxides.

Reactive oxygen species (ROS) represent a class of diverse active oxygen-containing substances, such as singlet oxygen (1O_2_), superoxide anions (O_2_^−^), hydroxyl radicals (OH·), hydrogen peroxide (H_2_O_2_), and nitric oxide (NO) [15]. ROS have been a companion of aerobic life for billions of years [16,17]. Originally, ROS were considered to be toxic metabolic by-products [18] that destroy cellular components. However, recent studies have shown that ROS serve as a double-edged sword in plant biology. ROS are indispensable during aerobic metabolism and play critical roles in signal transduction as essential regulators of many biosynthetic and metabolic pathways through transcriptional and post-translational modifications [19]. ROS generation and accumulation are involved in intracellular oxidative stress responses and plant development, which in turn regulate PCD. They strictly control cell proliferation and cell death through PCD, thereby maintaining a stable and balanced number of living cells. Although plant PCD signal transduction may comprise many parallel or crossed initial steps, ROS probably act as a common inducer and key molecular signal in plant PCD. Although many studies have demonstrated the importance of ROS in plant PCD, the types of ROS and the mechanisms through which they trigger dPCD remain unclear [20].

In the present review, we provide an overview of the most common instances of dPCD in plants and present information about their molecular regulation and diverse functions in plant reproductive and vegetative development. Additionally, we discuss the factors that regulate PCD, as well as the prominent roles of ROS in induction, signalling, and execution of plant cell death. We also highlight the sources of ROS and provide an overview of the mechanism of action of ROS in PCD regulation.

## 2. Sources of Intracellular ROS in Plants

Intracellular ROS are generated from many sources. Several organelles, including mitochondria, chloroplasts, endoplasmic reticulum (ER), and peroxisomes, produce ROS. In addition to these organelles, new sources of ROS, such as NADPH oxidases, amine oxidases, and peroxidases, have been identified in plants [15,21,22] (Figure 1).

In plant cells, mitochondria are considered to be the main source of ROS in non-photosynthetic tissues [22,23,24,25]. The mitochondrial electron transport chain (mETC) is responsible for generating ROS (mROS). ROS are produced at the inner mitochondrial membrane [26]. The key sites for mROS production are complex I (NADH- ubiquinone oxidoreductase) and complex III (ubiquinol: cytochrome c oxidoreductase; cytochrome bc1 complex) of the respiratory chain. During aerobic respiration, a small fraction of high-energy electrons leaks from complex I and complex III. Complexes I and III participate in the reversible conversion of hydroquinone into ubiquinone and generate a reaction intermediate, ubisemiquinone, which can accept electrons leaked from these two compounds. This process eventually leads to the reduction of the single electron of molecular oxygen, the formation of O_2_•^−^ with strong oxidation potential, and the generation of H_2_O_2_ and •OH through specific chemical reactions [27]. These mROS produced by mETC are released into the cytoplasm or mitochondrial matrix, which then play crucial roles in the signalling systems involved in energy metabolism and stress responses. mROS can lead to the induction of PCD in extreme cases.

In the presence of light, chloroplasts serve as the main source of ROS [28]. The oxygen-rich environment in chloroplasts leads to the production of ROS. Photosynthetic ETC located on the thylakoid membrane is the main pathway of ROS production. Two photosystem reaction centres (PSI and PSII) are considered the main sites for ROS production. At the acceptor site of PSI, H_2_O_2_ is formed mainly by the Mehler reaction in the presence of light [29,30,31]. At high concentrations, H_2_O_2_ can lead to PCD. H_2_O_2_ is the second messenger for signals generated by ROS [32]. In PSII, when the extent of light absorption exceeds the photosynthetic electron transport capacity, the number of quinone receptors of PSII decreases, and the probability of 1O_2_ production increases. The singlet-state oxygen (1O_2_) is excited by the reaction centre chlorophyll (Chl) of the triplet excited state [33]. The PSII core protein D1 is known as a prime target of 1O_2_ [34]. Some studies have shown that 1O_2_ is the signal molecule that regulates PCD in plants [35]. 1O_2_ also plays a crucial role in the transfer of signals from the chloroplast to the nucleus, which is mediated by the nucleus-encoded chloroplast-localised protein EXECUTER1 [15]. A study confirmed that the malate shuttle plays crucial roles in information transmission from the chloroplast to the mitochondria and ROS production in the mitochondria [28], indicating that information exchange occurs between the chloroplasts and mitochondria in plants.

Peroxisomes are the organelles that exhibit essentially the oxidative type of metabolism [36]. They are another important source of ROS and contain a complex antioxidant defence system that regulates the accumulation of ROS [37]. Large amounts of H_2_O_2_ are generated via the β-oxidation pathway in peroxisomes. Peroxisomes can produce H_2_O_2_ by converting glycolate into glyoxylate using the enzyme glycolate oxidase (GOX), mainly through photorespiration pathways [38]. Peroxisomal-sourced H_2_O_2_ can also trigger PCD through some metabolites [39].

In addition, ROS production is closely related to NADPH oxidase on the plasma membrane. NADPH oxidase catalyses O_2_ to form O_2_^−^ and then produces H_2_O_2_ through superoxide dismutase (SOD) [40,41]. In most cases, the lack of RBOH expression leads to extremely low levels of ROS production, resulting in the alteration of plant responses in terms of cell death and pathogen resistance.

Plant NADPH oxidases are known as respiratory burst oxidase homologues (RBOHs). The RBOH multigenic family comprises 10 members of *Arabidopsis* and 9 members of rice. The *Arabidopsis* robhD and robhF single and double mutants accumulate less H_2_O_2_, confirming that NADPH oxidase is required for H_2_O_2_ production [42].

## 3. Initiation, Execution, and Completion of PCD

In animal cells, PCD is divided into three stages: the initiation stage, the execution stage, and the degradation and scavenging stage. Plant PCD has also been reported to involve these three stages (Figure 2).

The initiation stage in animal and plant PCD is similar; it involves the production and transmission of several types of death signals that initiate intracellular death, such as activation of the death receptor, production of DNA damage stress signals, generation of survival signals, activation of apoptosis-related proteins by damage signals, and regulation of programmed death signals. Under normal circumstances, the process of PCD is kept under control by survival signals from the cellular environment and intracellular sentinel signals. Once a cell loses contact with its surroundings or undergoes irreparable internal cell damage, a cell death program is initiated. Cell death can also be initiated when cells receive conflicting signals that order them to divide and stop dividing simultaneously. Once PCD is initiated, the vacuolar membrane ruptures and releases various hydrolases [43]; therefore, the change in plant vacuolar membrane permeability acts as a marker of PCD initiation [44,45].

In animals, the execution phase involves the activation of caspases, which are a family of cysteine proteases. Caspases are of two types: initiators and executioners [46]. Initiator caspases that reside upstream of the caspase cascade activate executioner caspases, which then lyse the cellular substrates necessary for cell death in an orderly manner [47,48]. At least 14 types of caspases have been identified in animals; however, the isolation and purification of homologous caspases have not been reported in plants. However, many inhibitor experiments have shown the presence of caspase analogues, such as vacuole processing enzymes (VPE), metacaspases, saspase, and phytaspase, in plants [49,50,51]. A common feature shared by all caspase analogues is that they specifically break down peptide bonds located upstream of aspartic acid residues. Plant metacaspases are of two types. Type I metacaspases contain a domain with similarity to the zinc finger motif, and a proline-rich motif is also found in the LESION-SIMULATING DISEASE-1 (LSD1) protein, which is involved in the hypersensitive response (HR), a type of immune-response-associated PCD. Although type II metacaspases have no obvious functional domains, they determine the initiation of plant cell apoptosis. In rice (*Oryza sativa*), several caspase-like genes encoding metacaspases, such as *METACASPASE1-8* (*OsMC1-8*) [52] and VPEs (*OsVPE1-4*) [53,54], have been reported [55].

The execution stage is also associated with a change in mitochondrial permeability. Many studies have shown that the mitochondria can produce ROS and other signalling substances that can induce PCD; mitochondria can also sense and amplify death signals [28,56]. An adjustable permeability transition pore (PTP) is present at the junction of the inner and outer mitochondrial membranes that allows small molecules to pass freely. The opening of the PTP can cause the release of cytochrome C (Cyt C) and apoptosis-inducing factor (AIF). The released Cyt C then binds to pro-apoptotic apoptosis protease–activating factor 1 (Apaf-l), which itself binds to the caspase-9 precursor, leading to caspase-9 activation in the presence of ATP. Activated caspase-9 stimulates the caspase cascade, finally initiating PCD [57,58]. ROS, calcium ions (Ca^2+^), and caspase can all induce the opening of mitochondrial PTPs.

The degradation and scavenging stage involves the enzymatic hydrolysis of the death signal by caspases, fragmentation of chromosomal DNA, and phagocytosis of apoptotic bodies by phagocytes, eventually leading to cell death. Plant cells undergoing PCD share some of the morphological and biochemical characteristics with animal cells undergoing apoptosis, such as cytoplasmic concentration (manifested in plants as separation of plant cytoplasmic from the plant cell wall), nuclear and chromatin concentrations, DNA fragmentation, ROS accumulation, and Cyt C release [12,59,60]. A major difference between PCD in the animal and plant cells is that the outer surface of animal cells first blisters during apoptosis and is then swallowed by phagocytes; by contrast, plant cells undergo vacuolar autophagy and ablation of the cellular contents [61].

## 4. Mitochondria and Chloroplasts Perform Key Functions in PCD

Several organelles, such as mitochondria and chloroplasts, perform key functions in plant PCD. Mitochondria act as the centre of stress perception and regulate key physiological processes, including antioxidant defence response and cell apoptosis. Additionally, the mitochondria coordinate between external signals and intracellular development control systems [62]. The mitochondria also control multiple molecular cascades that lead to PCD and sense various stimuli of cellular stress. Mitochondria send out danger signals, causing steady-state changes in the environment of the cell or the whole organism, thereby promoting the intrinsic or systemic adaptive response of the cells.

Factors that induce apoptosis can cause uncoupling of the mitochondrial electron transport chain, ATP production, and changes in mitochondrial permeability, which in turn lead to a decrease in mitochondrial transmembrane potential (AWm) and the generation of harmful ROS [63,64]. The change in mitochondrial membrane permeability regulates the progression of PCD and determines cell survival. The mitochondrial morphology can remain normal in the early stages of PCD. The structure and function of the mitochondria significantly affect the process of pre-apoptosis [54,65,66]. With the progression of apoptosis, the mitochondria undergo several nonspecific changes, such as swelling and vacuolation. In *Arabidopsis* (*Arabidopsis thaliana*), ROS and heat shock treatment were reported to induce mitochondrial swelling and lead to cell death in leaves and protoplasts [67]. In another study, mitochondria in the developing endosperm of wheat became vacuolated after drought treatment [68].

Mounting evidence indicates that the chloroplasts also play a role in PCD. The chloroplasts, as an additional energy-transducing and ROS-generating compartment, are involved in the execution of PCD [69]. They induce PCD by generating ROS in the presence of light or even by Cyt *f* release [70,71]. Mitochondria and chloroplasts are not independent in the PCD process, and they may cooperatively execute PCD. Aken et al. described two models for PCD initiation by the mitochondria and chloroplasts, namely the serial model and the alternative model [20]. In the serial model, the mitochondria recruit chloroplasts as additional ROS producers in the presence of light, whereas in the alternative model, changes in ROS and Ca^2+^ affect both mitochondria and chloroplasts; thus, they jointly induce cell degradation. *Arabidopsis* is an example of the alternative model. MOSAIC DEATH 1 (MOD1) is an enoyl-ACP reductase that negatively regulates PCD [72]. *SOM3* (*SUPRESSOR OF mod1*) encodes a mitochondrial γ carbonic anhydrase, and *SOM42* encodes a mitochondrial-localised pentatricopeptide (PPR) protein. SOM3 and SOM42 can completely or partially restore the cell death phenotype and ROS accumulation seen in the *mod1* mutant. In the *mod1* mutant, a large amount of NADH accumulates in the chloroplasts, which generates malic acid and provides reducing power that enters the mitochondria through the malate-OAA shuttle, thereby initiating PCD [28]. This finding indicates that signalling between the chloroplasts and mitochondria plays a vital role in PCD regulation in plants [28,73].

## 5. Factors Regulating Plant PCD

Several lines of evidence indicate that similar signalling molecules are involved in both animal and plant PCD. Similar to animal PCD, these signals may converge into one or more common pathways in plants, which promote intracellular death and lead to common events such as DNA fragmentation. However, plant PCD signal transduction may comprise many parallel or crossed initial steps. Moreover, plant growth regulators, such as Ca^2+^ and ROS, may be the common regulatory and signalling factors involved in different PCD induction pathways.

Accumulating evidence points towards the role of phytohormones/plant growth regulators, such as ethylene, abscisic acid (ABA), jasmonic acid (JA), polyamines (PAs), and cytokinins (CKs), in the regulation of plant PCD [74]. Ethylene is crucial in triggering and promoting PCD [66]. Furthermore, leaves of the *Arabidopsis* ethylene-insensitive mutant *ethylene response 1* (*etr1-1*) live approximately 30% longer than wild-type leaves, as evidenced by delayed leaf senescence in the mutant background [75]. In Chinese cabbage, ethylene can upregulate the senescence-associated genes to promote PCD in papilla cells and break down the self-incompatibility mechanism [76,77]. Moreover, the treatment of flowers with ethylene resulted in carpel senescence and DNA fragmentation in pea (*Pisum sativum*) [78]. In the developing endosperm of the maize (*Zea mays*) *shrunken2* mutant (*sh2*), the ethylene content increased significantly, and PCD occurred earlier than that in the wild-type maize variety [79].

ABA is a positive regulator of the senescence process, and ABA levels increase in senescing leaves, which is accompanied by the increased expression of the genes encoding the key enzymes involved in ABA biosynthesis [80,81]. In maize, the appropriate onset and progression of PCD in the endosperm depend on the balance between ABA and ethylene; treating the wild type with an inhibitor of ABA biosynthesis phenocopied ABA-deficient mutants, including an increase in ethylene production and accelerated execution of PCD [82]. Abiotic stresses such as heat, cold, salt, and drought were reported to increase the ABA content in several different plants, resulting in ROS accumulation and induction of PCD [83,84].

Cytokinins (CKs) are considered to be the negative regulators of PCD. An increase in CK biosynthesis occurs only at the beginning of senescence, which leads to longer retention time of the green colour of leaves and delayed senescence of leaves and petals. Several studies have indicated that JA is a positive regulator of PCD, although not a major contributor [74]. Ca^2+^ is one of the core components of the eukaryotic signal transduction pathway. It acts as a second messenger in PCD and plays a role in PCD induction, initiation, and regulation. Ca^2+^ can activate PCD originating from the mitochondrial and ER signalling pathways. Ca^2+^ also activates calcium-dependent endonucleases to degrade DNA, and an increase in the Ca^2+^ concentration leads to PCD. An increase in the cytosolic Ca^2+^ concentration can also activate the cysteine protease calpain, thereby (re)activating caspase-like proteins to induce PCD [85].

Mitochondria-targeted quinones (SkQs) are also considered to be the negative regulators of PCD. SkQs are antioxidants that comprise plastoquinone or its derivative (antioxidant moiety), a cation that penetrates across membranes, and a hydrocarbon linker, for example, 10-(6’-plastoquinonyl) decyltriphenylphosphonium (SkQ1), 10-(6’-methylplastoquinonyl) decyltriphenylphosphonium (SkQ3), and 10-(6’-plastoquinonyl) decylrhodamine 19 (SkQR1) [86]. SkQs inhibit ROS generation and prevent the activation of protein kinases, thereby preventing PCD. SkQs prevent PCD, possibly through the following mechanisms: (1) accumulation of SkQs in the mitochondria reduces because of the decrease in Δψ, CN^−^; (2) the formation of mitochondrial ROS is inhibited due to the reduction in Δψ [87], and (3) the formation of ROS in the mitochondria can be inhibited by switching to cyanide-resistant respiration with alternative oxidase [86].

## 6. Role of Autophagy in Plant PCD

Cell death can be divided into developmental cell death, immunity-related cell death and disease-related cell death according to the cause of death [88]. Autophagy is one of the major cellular processes of degradation and recycling of cytoplasmic materials, including individual proteins, aggregates, and intracellular organelles, and it performs mainly survival functions in many eukaryotes, such as the regulation of innate immunity, stress adaptation, and PCD [88,89]. Research has indicated several complex functions of autophagy, including reproduction, development, primary metabolism, hormone signalling, cellular homeostasis, senescence, stress responses, and disease resistance during plants’ whole life [90,91].

Autophagy regulates different types of cell death processes through different pathways. During developmental cell death, autophagy plays a crucial role, based on the morphological characteristics. The functions of autophagy in the tapetal degradation during pollen maturation in rice and the cell death of suspensor cells during normal embyrogenesis in Norway spruce support this opinion [92,93], although limited supporting evidence is available. Some studies have shown that autophagy contributes to both the restriction and promotion of dPCD regulation by metacaspases. According to a study, autophagy promotes PCD in the tracheary element differentiation process in *Arabidopsis* [94].

Autophagy contributes to both death and survival functions in the regulation of R protein-mediated hypersensitive response (HR). The critical role of autophagy in the initiation and promotion of HR upon infection with avirulent strains of different pathogens in several plants (such as *Arabidopsis* and *Nicotiana benthamiana*) has been investigated [95,96]. Increasing facts have been gathered to clarify how autophagy contributes to disease-related cell death. AtBAG6 (BCL2-associated athanogene family protein 6) is the largest member of the Bcl-2-associated athanogene (BAG) family and was originally identified as a CaM-binding protein [46]. Li et al. found that AtBAG6 is proteolytically activated to trigger basal immunity and induce PCD, and activated AtBAG6 triggers autophagy in *Arabidopsis* [97].

## 7. Function of ROS in Plant PCD

Because ROS levels determine the cellular fate and signalling pathways, they are closely related to the regulation of developmental PCD and abiotic stress-induced PCD in plants. ROS can trigger multiple signalling pathways that lead to PCD and participate in the execution of PCD. For example, Levine et al. reported that treatment with protein biosynthesis inhibitors prevents H_2_O_2_-induced cell death [98]. Subsequent biochemical and genetic studies have clarified the key role of ROS in plant PCD. However, the mechanism of ROS action in PCD initiation remains unclear.

When the intracellular ROS level increases, ROS act as signalling molecules to activate the permeability transition channel at the mitochondrial membrane, resulting in the rapid release of Cyt C from the mitochondria, which marks the early stage of plant PCD. The release of Cyt C disrupts the electron transport chain, which leads to the production of high levels of ROS, causing a positive feedback loop that further strengthens the original death signal. The chloroplast can also sense and respond to environmental stresses, produce excessive ROS, expand response signals, and then aggravate the stress level [99,100]. This amplified death signal activates the transcription factors controlling PCD regulatory genes, and PCD defence genes may become suppressed.

Caspase-like proteases can further cleave and activate additional enzymes involved in cell death, such as nucleases and cell-wall-related enzymes. The activation of these enzymes leads to the appearance of tell-tale morphological signs of PCD, such as nuclear DNA degradation [101]. Over the course of plant PCD, two bursts of ROS are induced. The first peak may occur in the chloroplasts or leaves and lead to the influx of ROS-activated Ca^2+^, which causes the opening of mitochondrial PTPs. The second peak occurs when the electron transport chain is disrupted; however, the chloroplasts may also be involved in the light reaction [20].

Plant cells contain a regulatory network that balances redox metabolism by controlling the release of ROS, which would otherwise trigger signalling cascades and induce potentially harmful oxidation reactions [102]. During growth and development of plants under optimal conditions, the heart of each plant cell exhibits a dynamic balance between ROS production and the antioxidant defence system. Both growth and environmental stress can, however, cause an imbalance in ROS metabolism and lead to ROS accumulation [103]. Low concentrations of ROS, which act as signalling molecules, induce PCD in an orderly manner. The increase in ROS levels can disrupt the equilibrium, which is beneficial to the oxidative reaction and therefore produces oxidative stress. In this case, irreversible cell senescence and PCD occur [104,105]. In extreme cases, necrotic cell death caused by altered mitochondrial membrane permeability may occur when the antioxidant defence systems cannot accommodate extremely high concentrations of ROS (Figure 3).

High ROS levels during early anther development are required for tapetum cell degradation. The rapid reduction in ROS levels after the initiation of PCD is also essential for normal tapetum degradation, and the abnormal decrease in ROS levels leads to delayed PCD [106,107]. ROS promote PCD in barley (*Hordeum vulgare*) aleurone layers by inducing the secretion of hydrolases during seed germination [108]. The receptor kinase FERONIA (FER) controls the production of high ROS levels to induce pollen tube rupture [109].

## 8. Dynamic Changes in ROS Levels during PCD

Excessive accumulation of ROS can trigger PCD and play potential roles in inducing PCD. ROS burst occurs during the early stages of PCD, whereas during the late PCD stages, ROS production decreases to a low level [107]. Maintaining high ROS concentrations can lead to abnormalities in PCD, which in turn causes tapetal dysfunction and pollen abortion [110].

ROS exert a strong effect on PCD during the development of tapetum cells. The tapetum is a somatic helper tissue neighbouring microsporocytes and supporting gametogenesis. In numerous plants, pollen formation involves normal functioning and degeneration timing of the tapetum, with calcium and carbohydrates provided by the tapetum essential for male fertility. Both tapetum and the middle layer showed secretory activity, and both degenerated by programmed cell death (PCD) [105]. The specific spatiotemporal accumulation of ROS can regulate PCD in plant tapetum cells. During anther development, tapetum cells undergo PCD; otherwise, it can cause pollen abortion and lead to male sterility in plants. Several reports have suggested that ROS promote tapetal PCD at various stages during normal anther development, as the timing of ROS accumulation coincides with tapetal cell degradation [111]. In *Arabidopsis*, stage-specific expression of the NADPH oxidase RBOH, as mentioned earlier, may be involved in a temporal burst of ROS production. In *rboh* mutants, ROS contents decrease, which can affect the timing of tapetal PCD and lead to pollen abortion [110].

The connection between ROS and tapetum cell programmed death has been identified in multiple plant systems. In tobacco (*N. tabacum*) and tomato (*Solanum lycopersicum*), dynamic ROS levels are related to the initiation and progression of tapetal cell programmed death. The ROS levels during anther development in the pollen development stage were reported to decrease, resulting in partial male sterility [112]. In rice, *DEFECTIVE TAPETUM CELL DEATH 1 (DTC1)* controls the dynamics of ROS accumulation during pollen development. Downregulation of *DTC1* expression results in delayed tapetal cell degeneration and male sterility, indicating that *DTC1* regulates tapetal PCD by blocking the ROS-scavenging activity [113]. The rice MADS-box transcription factor MADS3 regulates tapetum development by modulating the expression of ROS production genes and maintaining the ROS balance. Tapetum development is abnormal in the *mads3* mutant with fully abortive pollens [107]. *WILD ABORTIVE 352* (*WA352*) is a mitochondrial gene expressed in the tapetum of the rice wild abortive cytoplasmic male sterile (CMS-WA) system. The mitochondrial protein WA352 binds to the mitochondrial protein CYTOCHROME OXIDASE 11 (COX11) encoded by a nuclear gene; the WA352–COX11 interaction prevents COX11 from scavenging ROS, resulting in early PCD of tapetum and pollen abortion [114]. In addition, rice *HEXOKINASE1 (HXK1)* controls appropriate ROS production and proper timing of tapetal PCD; *HXK1* expression is directly regulated by rice *ARGONAUTE 2 (AGO2*) through epigenetic regulation [106].

Although these data suggest that ROS are involved in the tapetum PCD in rice [107,114], some questions remain, such as whether these dynamic ROS changes in seed plant anthers are evolutionarily conserved, how ROS are produced in anthers, and how these dynamic ROS changes in anthers connect to an underlying regulatory network of transcription factors.

## 9. Discussion and Prospects

Since the introduction of the PCD concept, botanists have paid considerable attention to plant PCD. PCD occurs in an orderly manner, leading to cell death. PCD is essential for plant growth and plays a critical role in the development and metabolism of plants. Plant cells also undergo PCD in response to external stimuli to survive under adverse environmental conditions. However, plants have acquired the ability to adapt to their environment over the course of their evolution. Plant PCD involves a complex regulatory network that is regulated at the gene and protein levels by multiple signal transduction pathways. ROS, as the intracellular signalling molecules, are globally involved in regulating all aspects of life in most eukaryotes, and plants are no exception, as ROS influence plant growth, development, responses to external biotic and abiotic stimuli, and PCD.

The source, production mechanisms, and roles of ROS in plants have been identified. During normal cell metabolism, cellular ROS contents are maintained at a basal level in an equilibrium state. Maintenance of the ROS balance requires many genes, proteins, and other molecules that form a complex network. The accumulation of low ROS concentrations (but over basal levels) triggers PCD. When ROS concentrations increase, reversible cell senescence and PCD occur. The continuous increase in ROS eventually leads to cell death. These findings provide new clues to understand the importance of active oxygen function. However, the mechanism through which ROS participate in PCD through their effects on the intracellular environment and transcription status remains unclear. ROS act as signalling molecules in the regulation of plant PCD; however, research on the PCD regulatory networks is still in its infancy. Although the accumulation of ROS can lead to PCD, the final response to plant growth, development, and stresses are fundamentally different and should be investigated in further studies.

## Figures and Tables

**Figure 1 ijms-22-12942-f001:**
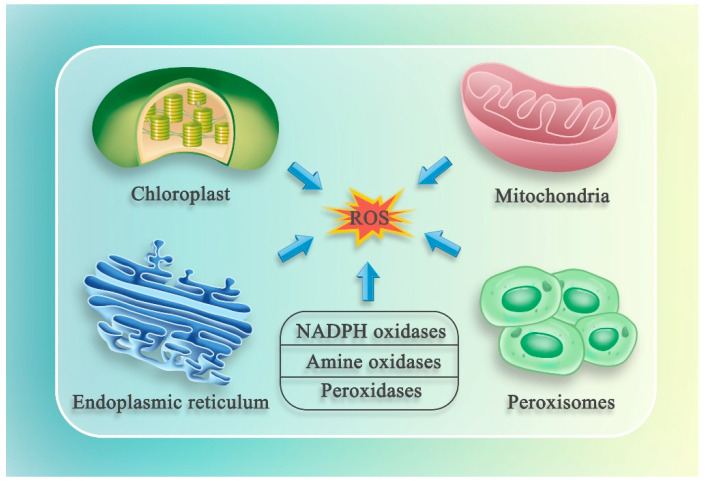
Diagram of the production of intracellular ROS. The mitochondria, chloroplasts, endoplasmic reticulum (ER), and peroxisomes are the main ROS-producing organelles. In addition to whole organelles, new sources of ROS have been found in plants, which include enzymes such as NADPH oxidases, amine oxidases, and peroxidases.

**Figure 2 ijms-22-12942-f002:**
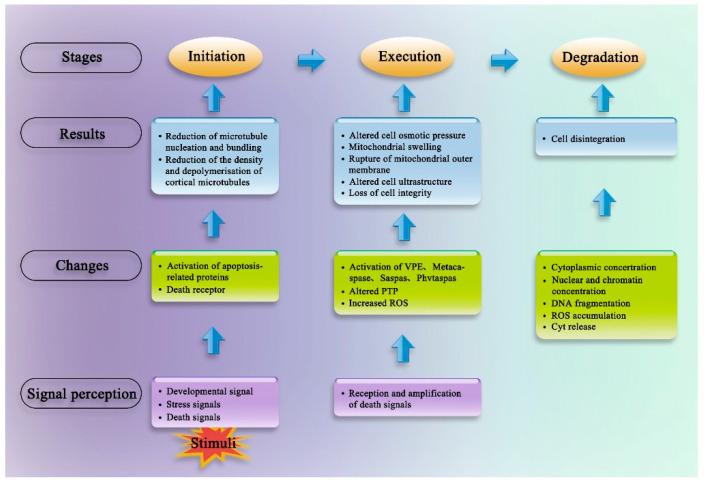
The three stages of animal PCD.

**Figure 3 ijms-22-12942-f003:**
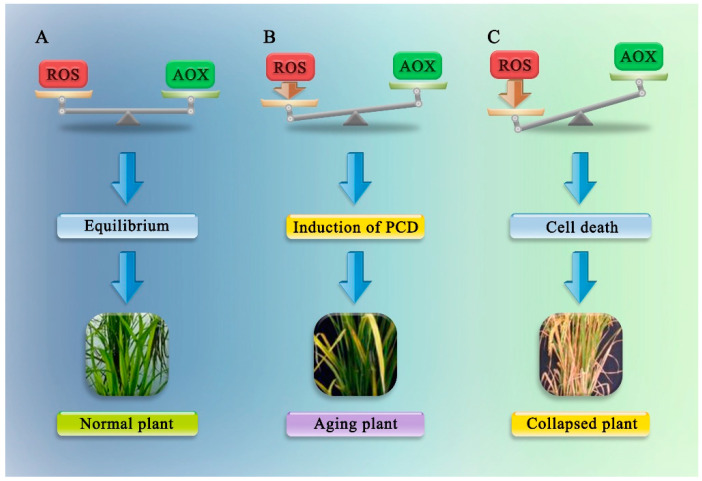
The regulatory network in plant cells that balances redox metabolism by controlling the release of ROS. Some disturbances in equilibrium lead to an increase in the intracellular ROS levels, which can cause damage to cells and induce cell death. (**A**) ROS = AOX, equilibrium; (**B**) excess ROS causes the induction of PCD; (**C**) excessive ROS levels cause cell death. AOX: antioxidants.

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
