# Peer review of "Initiation and Execution of Programmed Cell Death and Regulation of Reactive Oxygen Species in Plants"

_ijms, 2021, doi:10.3390/ijms222312942_

Round 1
Reviewer 1 Report
The authors provided a good overview of programmed cell death’s role and reactive oxygen species’ response in plants. Cell signal transduction and plant stress response were addressed. The role of mitochondria and chloroplasts in programmed cell death were described in detail. Diagrams and figures add value to the review.
It is suggested that the authors also add information to their manuscript and refer to recent reviews in their reference list namely the review written by Samuilov VD, et al. Mitochondrion. 2019. PMID: 29723685 and Kurusu T, et al. J Plant Res. 2017. PMID: 28364377.
Author Response
Response to Reviewer 1 Comments
point 1:It is suggested that the authors also add information to their manuscript and refer to recent reviews in their reference list namely the review written by Samuilov VD, et al. Mitochondrion. 2019. PMID: 29723685 and Kurusu T, et al. J Plant Res. 2017. PMID: 28364377.
Response 1: Thank you for your suggestion. We have referred to the suggested reviews and added relevant points to our article. Based on these references, we have tried to add information about the Mitochondria-targeted quinones (SkQs) and autophagy in plant PCD. Please refer to the last paragraph under the subheadings "5. Factors Regulating Plant PCD" and "6. Role of Autophagy in Plant PCD".
Reviewer 2 Report
In this review article, the authors discussed the PCD mechanisms in plants. The review is interesting but needs improvements:
1-The author should clarify the basic fundamental differences between PCD in plant cell and animal cell. Also basic differences between plant cell and animal cell.
2-Tables are required to show up-to-date information in PCD in plants under both normal and pathological conditions
3-A paragraph showing the role of autophagy in plants as antiapoptotic mechanism should be added
4-In Fig.3, do not really understand the differences between cell death in aging plant and collapsed one. What is the signaling apoptotic pathways in both? Are they same?
5-The authors should define and mention the morphology and function of tabetum cell.
6- The paper has some grammar and language mistakes and should be checked by native English speakers
